# Investigating the Use of Recycled Pork Fat-Based Biodiesel in Aviation Turbo Engines

**Grigore Cican [1],\*, Marius Deaconu [2], Radu Mirea [2], Laurentiu Ceatra [2], Mihaiella Cretu [2] and Tănase Dobre [3]**

1.   Faculty of Aerospace Engineering, Polytechnic University of Bucharest, 1–7 Polizu Street, 1, 011061 Bucharest, Romania

2.   National Research and Development Institute for Gas Turbines COMOTI, 220D Iuliu Maniu, 061126 Bucharest, Romania; marius.deaconu@comoti.ro (M.D.); radu.mirea@comoti.ro (R.M.); laurentiu.ceatra@comoti.ro (L.C.); mihaela.cretu@comoti.ro (M.C.)

3.   Faculty of Applied Chemistry and Materials Science, Polytechnic University of Bucharest, 1–7 Polizu Street, 1, 011061 Bucharest, Romania; tanase.dobre@upb.ro

\*   Correspondence: grigore.cican@upb.ro

**Abstract:** This paper presents an analysis of the possibility of using recycled pork fat-based biodiesel as fuel for aviation turbo-engines. The analysis consists of the assessment of four blends of Jet A kerosene with 10%, 30%, 50%, and 100% biodiesel and pure Jet A that was used as reference in the study. The first part of the paper presents the physical-chemical properties of the blends: density, viscosity, flash point, freezing point, and calorific power. Through Fourier transform infrared spectroscopy (FTIR) analysis, a benchmark was performed on the mixtures of Jet A with 10%, 20%, 30%, 50%, and 100% biodiesel compared with Jet A. The second part of the paper presents the test results of these blends used for fuelling a Jet Cat P80 turbo engine at the Turbo Engines Laboratory of the Aerospace Engineering Faculty of Polyethnic University of Bucharest. These functional tests were performed using different operating regimes as follows: idle, cruise, intermediate, and maximum. For each regime, a testing period of around 1 min was selected and the engine parameters were monitored during the test execution. The burning efficiency was calculated for the maximum regime for all mixtures. To evaluate the functioning stability of the turbo engine using biodiesel, two accelerometers were mounted on the engine support that recorded the radial and axial vibrations. Moreover, to assess the burning stability and to identify other acoustic spectral components when biodiesel is used, two microphones were placed near the jet region. A comparative analysis between blends was made by taking the Jet A fuel as reference.

**Keywords:** biodiesel; recycled pork fat; turbo engine performance; bio-aviation fuel

## 1. Introduction

The researchers from the aviation field are trying continuously to reduce aviation engines' emissions and, in "Flightpath 2050", it is envisioned to achieve a considerable decrease by 2050, up to 75% of $CO_2$ emissions and 90% of $NO_x$ emissions per passenger kilometer, these are relative to the typical aircrafts in 2000 [1]. For $CO_2$ and $NO_X$ emissions, an increase of 21% and 16%, respectively, is predicted by 2040, according to the European Aviation Environmental Report from 2019 [2].

According to International Civil Aviation Organisation's (ICAO) 2050 Vision for Sustainable Aviation Fuels, the aviation sector does not have other liquid fuel options as a source of energy, unlike other sectors such as road transport, and hence calls for a 'significant proportion of conventional aviation fuels to be substituted with sustainable aviation fuels by 2050' [3,4].

Experts in natural resources state that, in the next 60 years, natural resources of petroleum and natural gas will be depleted if the same consumption rate is maintained [5]. Fossil fuels are the base of socio-economic development, but they always have a negative impact on the environment, so bio-fuels are the most feasible alternative sources of fuel for turboengines [6].

Bio-fuels have been and are used in many engineering fields, where there are applied in thermal engines and more precisely for the internal combustion engines. Agarwal et al. [7] present a critical appreciation of the effects of biodiesel versus conventional diesel in terms of engine performances, emissions, and combustion, but they do not take into account the use of combustible blends that contain classical fuels (e.g., diesel or kerosene) and fuels obtained from vegetal oils/animal fats (biodiesel). Other research studies have focusing on the use of bio-fuel/biodiesel in internal combustion engines are presented in [8–10].

Over time, there have been many studies conducted to evaluate alternative fuels used to power turbine engines. Laboratory tests and tests on in-flight aviation turbine engines have been performed.

There are many studies in which biodiesel has been used as fuel for turbo engines. French K.'s paper [11] evaluated the performance of an SR-30 turbojet gas turbine engine powered by canola oil biodiesel. Habib et al. [12] has tested different kinds of blends ranging from 50% to 100% (B50 and B100) volumetric blends of biodiesel and Jet A-1. The main conclusions of the study are that the ratio of thrust/specific fuel consumption was similar to that in the results of Jet A-1, the admission temperature was higher overall for biodiesel/Jet A-1 mixtures, and the temperature of the exhausted gases was not significantly different. Chiang et al. [13] have examined a 150 kW Teledyne RGT-3600 turbo engine fuelled with mixtures consisting of 10%, 20%, and 30% biodiesel. Krishna [14] has tested 20%, 50%, and 100% soy-based biodiesel mixtures in a 30 kW capstone CR30 gas-fired turbo. In 2010, another study [15] was performed that aimed to determine and to quantify the emissions from the ignition of different biodiesel blends and pure biodiesel using an auxiliary power unit. Other studies related to the efficiency of biodiesel can be found in [15–19].

Another study [20,21] presents the evaluation of the emissions and combustion of bio-fuel blends within a gas turbine combustor. Over time, there have been several demonstrative tests. Demonstration flights have been conducted to examine the maintenance conditions of a drop-in fuel working in real conditions. Several aircraft types as such Boeing 787, Boeing 737–800, Boeing 747–400, Bombardiere Q400, Airbus A321, and Falcon 20, respectively, have performed demonstrative flights being fed with different combustible Jet A/bio-fuel blends [22–32].

It is well known that the use of the bio-fuels in aviation engines does not alone contribute to the emission control of $CO_2$, but to particulate matter reduction. Thus, Moore et al. [33] have performed a study within this direction by sampling a DC-8 aircraft burning Jet A fuel and a mixture of Jet A with Camelina-based bio-fuel. The works of [34] and [35] present the perspectives and challenges when biomass-derived fuels are used for aviation.

Because of imminent fossil fuel depletion, animal fats and vegetable oils are envisaged to be valuable renewable resources used in biodiesel manufacturing, as their continuous generation is creating waste management problems. It has been assessed that the use of the management methods for traditional waste (e.g., cremation, dumpsite) discharges greenhouse emissions, with a significant impact on global warming.

The paper presents an analysis of the possibility of using biodiesel from the recycled pork fat as fuel for the aviation turbo-engines.

An analysis of some mixtures of kerosene (Ke) fuel with biodiesel (BD), (Ke) with 10% (BD), 30% (BD), and 50% (BD) was made. After determining the physical-chemical properties of the mixtures, a measurement campaign followed where burning tests were done on the Jet Cat P80 micro-turbo engine.

## 2. Feedstock and Methods for Bio-Fuel

Biodiesel can be obtained through various processes and technologies [36]. Generally, the main raw material used for producing biodiesel can be categorized as follows: oils with vegetable provenience, animal fats, used cooking oils/fats, and algae.

Bio-fuels can also be categorized as follows: biodiesel, crude bio-oil, bio ethanol, biogas, and bio hydrogen, highlighting that the liquid fuels are frequently used in the implementation of conventional engine infrastructures.

The conversion of the feedstock into bio-fuel usable for turbo engines can be accomplished through several conversion paths: gas to jet, oil to jet, sugar to jet, and alcohol to jet [37–39]. The above conversion technologies are extensively used in the current industry, specifically the oil-to-jet path, which implies hydro processing [40]. The molecular structural modification and deoxygenation are important actions before actually applying the triglycerides in aviation bio-fuels [41]. Producing biodiesel from waste animal fats has an important capability as this raw material does not challenge the food industry and has a great input for the global reduction of wastes. Grease, fat, and oil are lipid-rich waste resulting from different origins including restaurants, hotels, food industries, and families that use them daily, so they are considered a reliable raw material for biodiesel processing. The possibility of using fats and oils as raw materials for producing biodiesel is investigated in [42]. An evaluation on the use of spirulina, waste cooking, and animal fats in diesel motors is presented in [43].

The successful manufacturing of biodiesel from different animal fats has been described by different groups [44,45]. Biodiesel manufacturing from pork fat using the trans-esterification method is presented in [46].

The feedstock type as well as the presence of impurities exert a tremendous influence on the quality of the resulting biodiesel. A high concentration of glycerol or/and glycerides in biodiesel results in a negative influence on the quality of fuel and can in general reduce the durability of the engine, so it becomes compulsory for crude biodiesel to be purified [47,48].

To prove the commercial usefulness of the biodiesel, another study [49] presents an optimization model used in the manufacturing of animal fat-based biodiesel and supply, which lowers, on one hand, the total cost of biodiesel supply and, on the other hand, carbon emission during operations.

## 3. Standards, Compatibility, and Characteristics

In the future, bio-fuels will be adjusted to meet the requirements related to execution and the physical characteristics of fossil fuels [50]. Their main properties are presented in [51,52]. The features of biodiesel generated from different sources of oil, that is, pork fat, as well as the comparisons with international standards, are presented in [53].

Jet-A and Jet-A1 are identical apart from the freezing point parameter. Jet-A is the U.S. specification, while Jet-A1 is the guideline used in Europe. For this reason, Jet-A and Jet-A1 will be presented in more detail for the objective of the article. The specifications for Jet-A fuel used in aviation gas turbines and biodiesel are not exactly similar, as presented by ASTM D1655 [54] and ASTM D6751 [55]. One of the main concerns regarding jet fuels is related to their capacity to function at very low temperatures. Considering this, a freezing point below −40 °C is required for Jet-A fuel, while an even lower freezing point (−47 °C) is needed for Jet-A1. The standard values for the cloud point (the point at which waxes within the fuel solidify, modifying the form of the fuel and growing its resistance to movement through-filters and pumps) of biodiesel are much higher, in the range of −25 to 26 °C. This implies that operating an aviation gas turbine on pure biodiesel (B100) is not a practical recommendation. Nevertheless, the analysis has concluded that combinations up to 20% (B20) can be utilized while following the ASTM D1655 specifications and needing slight or no changes to a gas turbine and its fuel system [56].

## 4. Test Bench and Experimental Procedure for Turbo Engine

This chapter presents the test bench used for the evaluation of the biodiesel blend and the turbo engine performances.

The experiments were performed in collaboration with INCDT COMOTI on a Jet CAT P80® turbo engine [57], presented in Figure 1, which is part of the Turbo Engines Laboratory of the Aerospace Engineering Faculty of Polyethnic University of Bucharest.

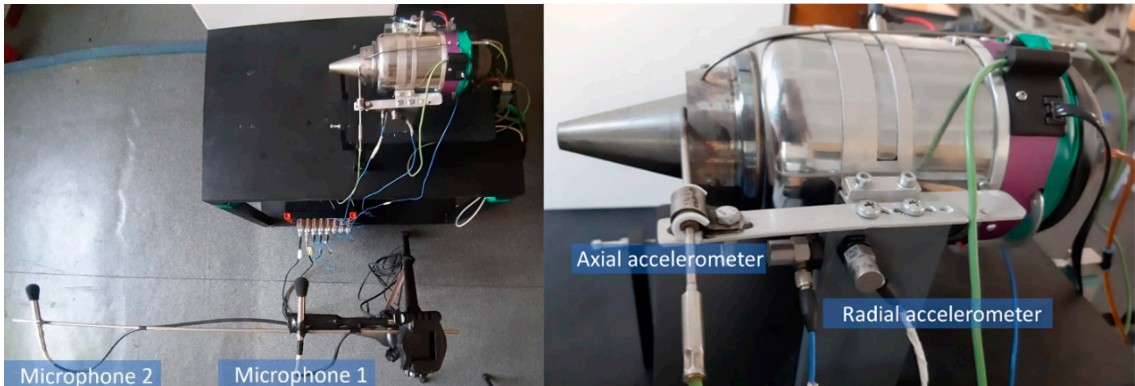

**Figure 1.** Test bench instrumentation (microphones and accelerometers location).

The studied blends in this paper are as follows: kerosene fuel (Ke) with 10% (BD), 30% (BD), and 50% (BD). In each blend, 5% of Aeroshell 500 oil was inserted for engine lubrification.

In terms of turbo engine parameters and performances, the instrumentation records the temperature in front of the turbine and after compressor, as well as the thrust force, the pressure in the combustion chamber, and the fuel and air flow.

The engine runs by keeping a constant speed law of the shaft, which is not changed by the fuel used, and to maintain the shaft speed, the fuel will be variously introduced in the combustion chamber. Considering that the compressor will have the same speed for all mixtures of fuel, the same pressure and the air flow will be generated.

By keeping the shaft speed constant, several parameters can be assessed: fuel flow (Qc), temperature in front of the turbine ($T_3$), and thrust (F), These parameters are analysed when the engine is running in idling regime (meaning 18.7% of the throttle gas), at cruise (meaning 30% of the throttle gas), at a selected regime (60% of the throttle gas), and at the max regime (94% of the throttle gas for the safety functioning condition). These regimes are kept for 1 min and then they were stabilized.

During the entire functioning period, vibrations were monitored on both the axial and radial directions, using two accelerometers, while the noise was monitored with two microphones; all the monitoring instruments are depicted in Figure 1. The vibration and acoustic measurements were performed with a Sirius multichannel acquisition system produced by Dewesoft, Trbovlje, Slovenia two PCB 352C03 accelerometers, and two microphones 40AE with 26AC preamplifiers produced by GRAS, Holte, Denmark. The microphones were placed at a distance of 0.6 m from the engine and at a distance of 0.55 m between them. Two accelerometers were rigidly mounted on the engine support on two axes, axial and radial, as presented in Figure 1. Through these measurements, we followed the mechanical and thermo-acoustic behaviour of the engine and stability during the testing of the blends.

## 5. Analysis of Physical-Chemical Properties for Fuel Blends

In this chapter, the main physical-chemical properties determined for the fuel blends/mixtures used in the tests and the equipment used for determinations are presented.

### 5.1. Density of the Fuel Determination

Fuels' density is experimentally determined by means of a thermo-dens meter, according to standard SR EN ISO 3675/2002 [58]. The equipment used to perform this assessment is depicted in Figure 2.

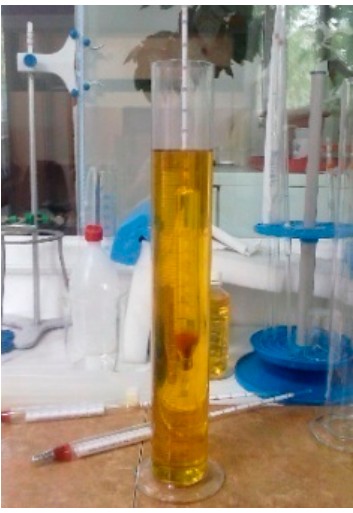

**Figure 2.** Fuels' density measurement.

By means of the tables in the standard, the measured values are converted into density values at 15 °C or 20 °C.

### 5.2. Flash Point Determination

The flash point represents the lowest temperature of a sample that forms a flammable blend with air under monitored conditions. The parameter is used to assess the general flammability hazard of a material. The equipment used for flash point determination is an Automatic Flash Point Tester Cleveland, produced by Scavini, Banevo, Italy based on the ASTM D92 test method [59]. The sample is firstly homogenized and then the required quantity is transferred to the test cup. The temperature of the sample rises rapidly at first and subsequently at a slow, continuous rate, as the temperature approaches the flash point set value. The flash point is determined when the vapours above the sample ignite when applying a test flame. The equipment used for performing this assessment is depicted in Figure 3.

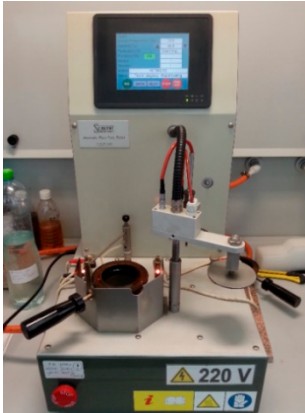

**Figure 3.** Automatic Flash Point Tester Cleveland.

### 5.3. Kinematic Viscosity Determination

The kinematic viscosity is experimentally determined, according to SR EN ISO 3104/2002 [60], using a capillary viscometer kept in a bath provided with a mechanical stirrer and thermostatic control. This approach consists of the determination of the necessary time for a known volume of sample to flow through a standardized capillary tube. The kinematic viscosity is calculated by multiplying the measured time (in seconds) with the capillary constant (which differs from one capillary to another), with the measurement unit as $mm^2/s$ (1 $mm^2/s$ = 1 cSt). The equipment used for performing this assessment is depicted in Figure 4.

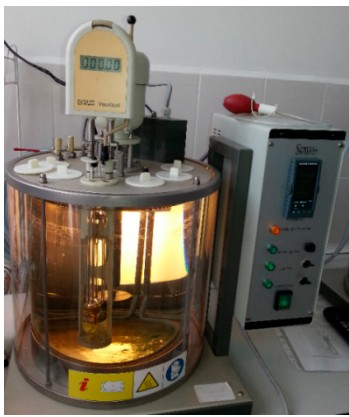

**Figure 4.** Kinematic viscosity determination equipment.

### 5.4. Calorific Power Determination

The calorific value of petroleum products is determined in accordance with ASTM D240-17 [61] "Standard test method for Heat of Combustion of Liquid Hydrocarbon Fuels by Bomb Calorimeter". The equipment used in determinations is the IKA WERKE C 2000 isothermal calorimeter provided with the C 5012 calorimeter bomb. The equipment used for performing this assessment is depicted in Figure 5.

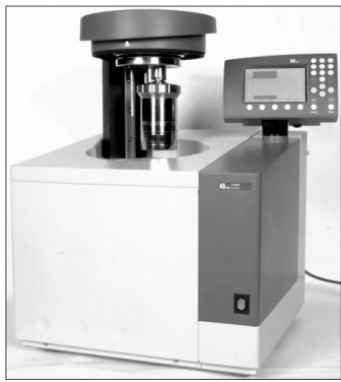

**Figure 5.** IKA WERKE C 2000 Calorimeter.

The calorimetric tests are performed in the measuring cell, and consist of the combustion of the fuel sample under precisely stated conditions. For this purpose, the measuring cell consists of the following components: an inner vessel provided with water filled jacket (outer vessel), a stirrer for achieving a uniform heat distribution in the inner vessel, and a water circulation system provided with a heating element for both regulating the system temperature and automatically filling the inner vessel.

A weighted sample is introduced into the combustion vessel, where it is then burned, while the temperature increase of the calorimetric system is measured.

The specific calorific value of the sample was automatically computed taking into account the sample mass, the calorific value of the calorimetric system and the increase of the water temperature in the inner vessel of the measuring cell.

The IKA WERKE C 2000 type calorimeter produced by IKA Analisentechnik GmbH, Germany displays the value of the upper calorific value of the analysed sample, which is expressed in kcal/kg or MJ/kg. The equipment is also provided with a software that allows the automatic calculation of the upper calorific value to the state for analysis.

During the combustion of the sample, the formed water vapours are exhausted together with the flue gases, without returning the corresponding heat of evaporation, thus the liquid is actually characterized by the lower calorific power.

*5.5. Freezing Point Determination*

The working procedure respects the requirements specified in SR EN ISO 3013/1997, "Liquid petroleum products—Determination of the freezing point".

The equipment used for these determinations is described in Figure 6 and consists of the following:

(1) 20 ± 2 mm in diameter glass tube provided with a perforated cork, used for the insertion of a thermometer when performing the determinations and an indicative mark, used to achieve the filling of the test tube with an adequate volume of sample;

(2) 40 ± 2 mm in diameter glass tube in which the glass test tube (1) is fixed by the means of a perforated cork plate (3);

(4) thermally insulated cooling bath, equipped with a support (5) for both the glass tube (2) and the thermometer (7);

A mercury thermometer (6) and an alcohol thermometer (7), respectively.

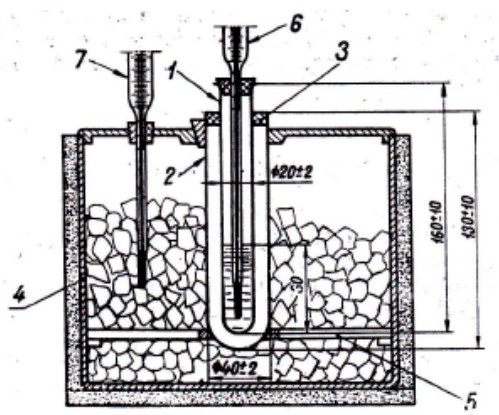

**Figure 6.** Equipment for freezing point determination.

Depending on the temperature that must be attained, the cooling mixture can be prepared from the following:

A mixture consisting of water and ice is used when temperatures higher than 0 °C must be attained;

A mixture consisting of ice and salt is used when temperatures in the range −15 °C ÷ 0 °C must be attained;

A mixture consisting of denatured alcohol/gasoline and liquid carbon dioxide/liquid propane is used when temperatures lower than −15 °C must be attained.

An adequate amount of sample is gradually cooled and the freezing feature is examined at each 3 °C. The lowest temperature, at which the sample surface remains still, is considered the freezing point. It must be mentioned that the equipment allows the determination of a freezing point as low as −35 °C.

*5.6. FTIR Analysis (Fourier Transform Infrared Spectroscopy)*

FTIR analysis (Fourier transform infrared spectroscopy) of the fuel blends consisting of Jet A kerosene and 10%, 20%, 30%, and 50% biodiesel, considering Jet A kerosene spectra and 100% biodiesel spectra, respectively, as reference, was performed.

The equipment used for performing this assessment is depicted in Figure 7.

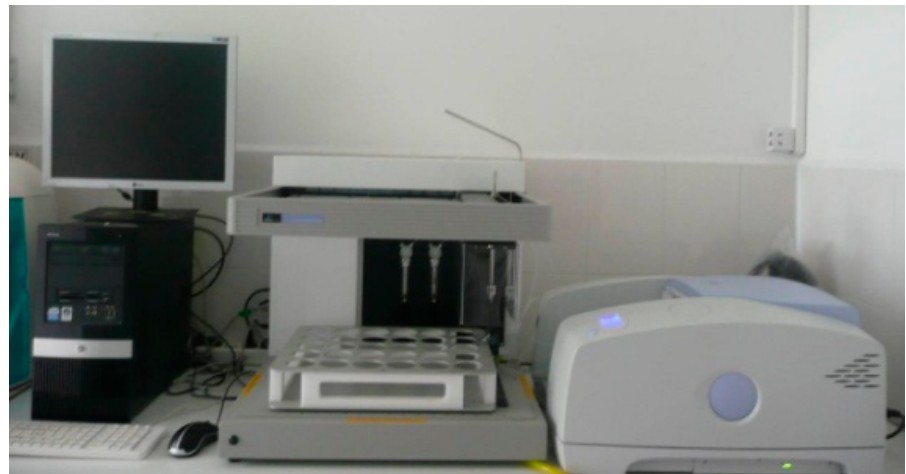

**Figure 7.** Fourier transform infrared (FTIR) Spectrum OilExpress Series 100, v 3.0 spectrometer.

## 6. Experimental Results

The results are presented considering the two directions mentioned above.

*6.1. Physico-Chemical Properties for Fuel Blends Experimental Results*

Following the determination of the physico-chemical properties of the analyzed fuel blends, the measured values are tabulated and centralized in Table 1.

**Table 1.** Experimental results for physical-chemical properties for fuel blends. BD, biodiesel.

| Sample | Flash Point °C | Viscosity at 40 °C cSt | Density at 22 °C g/cm$^3$ | Freezing Point °C | Higher Calorific Value kJ/kg |
|---|---|---|---|---|---|
| Ke + 5% Aeroshell | 42.3 | 1.39 | 0.817 | <−35 °C | 45.292 |
| Ke + 10% BD pork | 44.2 | 1.51 | 0.823 | <−35 °C | 44.403 |
| Ke + 20% BD pork | 50.2 | 1.82 | 0.830 | <−35 °C | 43.67 |
| Ke + 30% BD pork | 54.7 | 2.06 | 0.836 | −29 °C | 43.302 |
| Ke + 50% BD pork | 57 | 2.62 | 0.850 | −23 °C | 41.997 |
| 100% BD pork | 161 | 5.08 | 0.875 | −6 °C | 39.323 |

Analysing the data from Table 1, it can be observed that the freezing point of the combustible blends increases slightly as the ratio of biodiesel in the blends increases, while the upper calorific power of the combustible blends decreases slightly when the ratio of the biodiesel in the blends grows.

Consequently, it can be concluded that an increase in the percentage of biodiesel in the formulated tested fuel blends results in an increase of the freezing point and, implicitly, the flight will have to be performed at a much lower altitude than in the case of the standard flying altitude that resides from the use of a Jet A fuel type, as a reference. Besides, the increase of the percentage of biodiesel in the formulated tested fuel blends results in a decrease of the upper calorific power and, consequently, a much larger amount of fuel will be required than in the case of a Jet A fuel type.

The FTIR spectra highlighted in different colours in Figure 8 are attributed to the fuels used in the tests performed as follows.

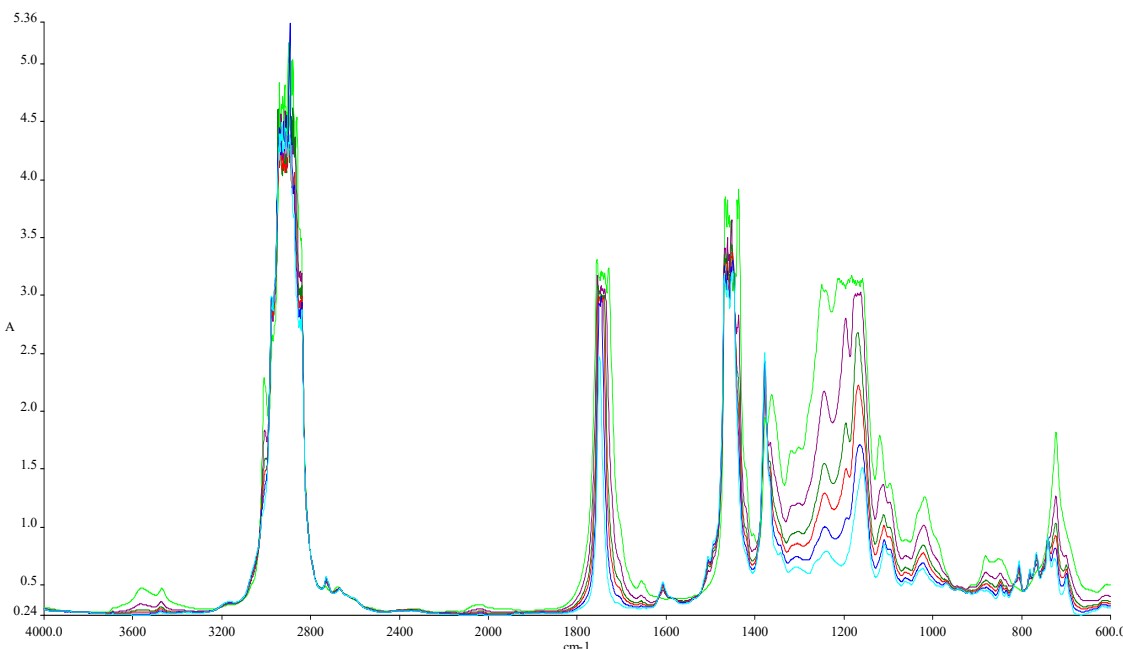

**Figure 8.** FTIR spectra of the blends (spectra of 100% biodiesel (BD) fuel—light green, spectra of 50% BD fuel—purple, spectra of 30% BD fuel—dark green, spectra of 20% BD fuel—red, spectra of 10% BD fuel—blue, and spectra of Jet A fuel (Kerosene) with 5% Aeroshell 500 turbine engine oil added—light blue).

When the FTIR spectra are compared, variations appear at 1745.83 cm$^{-1}$ (C=O stretching), 1030.98 cm$^{-1}$, 1117.54 cm$^{-1}$, and 1170.23 cm$^{-1}$ (C–O alkoxyl stretching), but their intensity differs according to the concentration of biodiesel, as shown in Figure 8. These peaks increase as biodiesel concentration increases. Fatty acid methyl esters (FAMEs) are a sign of the amount of the biodiesel present in each of the blends, as FAMEs appear at 1745.83 cm$^{-1}$ and 1170.23 cm$^{-1}$–1030.98 cm$^{-1}$. Methyl esters also show their absorptions characteristics (A = absorbance) in the peak around 1820–1680 cm$^{-1}$ [62].

### 6.2. Micro-Turbojet Engine Experimental Results

Table 2 shows the main monitored parameters of interest in the performance analysis that were recorded during the operation of the turbo engine. The data were averaged for 1 min of functioning at each regime. The monitored and recorded parameters are as follows: fuel flow, thrust, gas temperature in front of the turbine, and axial and radial vibrations.

Analysing the data from Table 2, the first conclusion that can be drawn is that the integrity and functionality of the turbo engine are not affected when biodiesel is added in fuel.

When the turbo engine operates at idling regime, the temperature in front of the turbine has the greatest fluctuations of all operating regimes. Fuel consumption has small fluctuations for all four operating regimes and for all five fuel blends.

Thrust exhibited positive variations of few percentage when the turbo engine worked at idle and cruise regimes.

Regarding the vibrations, the use of Jet A fuel/biodiesel pork fat-based blends for powering the turbo engine did not induce significant levels on the radial direction.

The values tabulated and centralized in Table 2 are graphically presented in the form of charts in Figures 9–16, in order to achieve a more detailed visualization of the fluctuations revealed by the monitored and recorded parameters.

**Table 2.** Main turbojet engine measured parameters.

| Regime | Blend | $T_3$ (°C) | Qc (L/h) | F (N) | Acc-Radial (mm/s) | Acc-Axial (mm/s) |
|---|---|---|---|---|---|---|
| Regime 1—idle 18.7% | Ke | 651 | 7.1 | 4.15 | 1.9 | 1.5 |
| | Ke + 10%BD | 656 | 6.9 | 4.38 | 0.6 | 1.2 |
| | Ke + 20%BD | 665 | 6.9 | 4.42 | 0.7 | 1.3 |
| | Ke + 30%BD | 668 | 6.8 | 4.33 | 0.8 | 1.3 |
| | Ke + 50%BD | 705 | 6.6 | 4.40 | 0.5 | 1.3 |
| Regime 2—cruise 30% | Ke | 648 | 10.3 | 12.26 | 1.2 | 1.1 |
| | Ke + 10% BD | 643 | 10.3 | 12.51 | 0.9 | 1.5 |
| | Ke + 20% BD | 637 | 10.4 | 12.58 | 0.9 | 1.4 |
| | Ke + 30% BD | 635 | 10.5 | 12.60 | 0.8 | 1.8 |
| | Ke + 50% BD | 615 | 10.2 | 12.61 | 0.8 | 1.6 |
| Regime 3—intermediate 60% | Ke | 609 | 16.8 | 39.3 | 2.3 | 3.1 |
| | Ke + 10% BD | 608 | 16.7 | 38.8 | 2.8 | 3.7 |
| | Ke + 20% BD | 602 | 16.6 | 38.9 | 4 | 4.7 |
| | Ke + 30% BD | 596 | 16.7 | 39.1 | 3.5 | 4.2 |
| | Ke + 50% BD | 593 | 16.8 | 39.3 | 3.4 | 4.5 |
| Regime 4—maxim 94% | Ke | 714 | 22.7 | 72.5 | 2.5 | 2.2 |
| | Ke + 10% BD | 714 | 22.5 | 72.0 | 1.9 | 1.8 |
| | Ke + 20% BD | 713 | 22.7 | 72.0 | 1.8 | 1.8 |
| | Ke + 30% BD | 714 | 22.7 | 71.7 | 1.7 | 1.7 |
| | Ke + 50% BD | 704 | 22.7 | 71.9 | 1.8 | 1.8 |

Figure 9 provides the charts that describe variation of the temperature in front of the turbine for the four operating regimes and for the five different types of fuel.

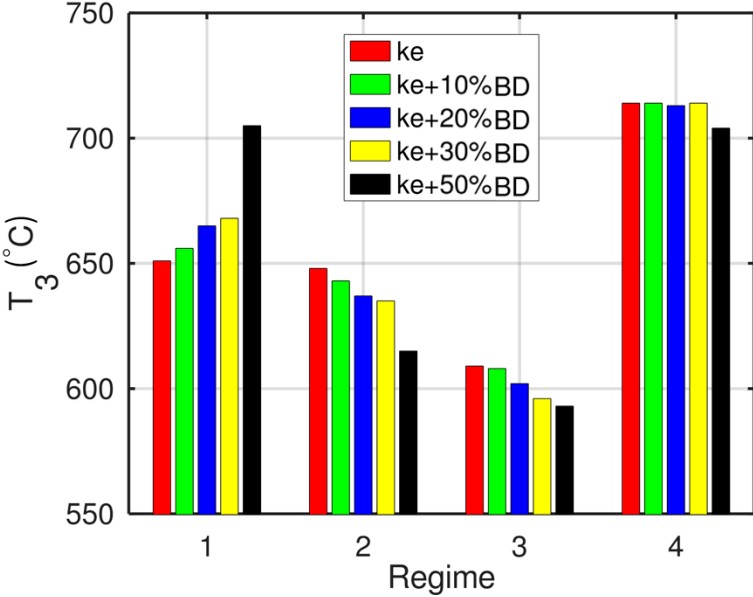

**Figure 9.** Variation of $T_3$(°C) depending on regime and blends.

From Figure 9, it can be noticed that the temperature measured in the combustion chamber is higher when the turbo engine is powered by each of the four tested biodiesel blends while operating in the idle mode, compared with the case of using a Jet A type fuel, without exceeding the upper prescribed limit for engine exploiting of 800 °C. In both the cruise and intermediate operating regimes, the temperature measured in the combustion chamber decreases below the temperature attained in the combustion chamber when using a Jet A type fuel, considering the reference temperature, when the

biodiesel concentration in the four tested combustible blends increases. In the maximum operating regime, the temperature measured in front of the turbine when using the four combustible blends exhibits small fluctuations against the temperature attained in the combustion chamber when using a Jet A type fuel. These small fluctuations can be attributed to the reading errors of the used thermocouple.

Figure 10 provides charts showing the variation of fuel flow of the five fuel blends tested for the four operating regimes. As can be observed, there are no notable fluctuations in the fuel flow when the turbo engine is operated.

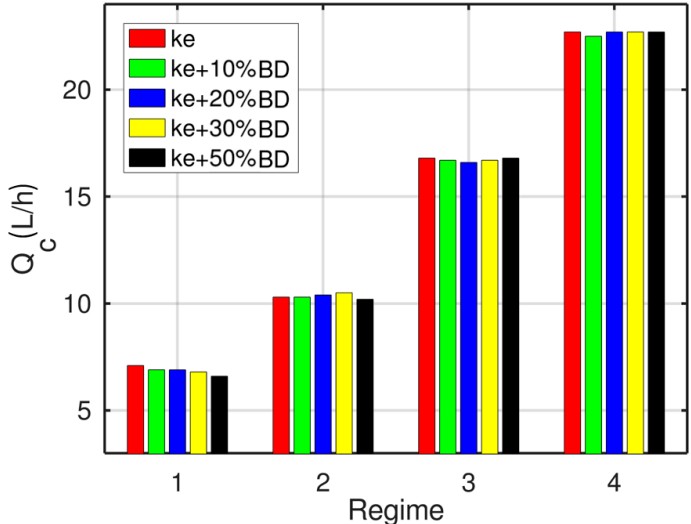

**Figure 10.** Variation of Qc (L/h) depending on the regime and blend.

Figure 11 provides charts displaying the variation of the thrust when the turbo engine is powered by the five fuel blends while operating in the four regimes mentioned above.

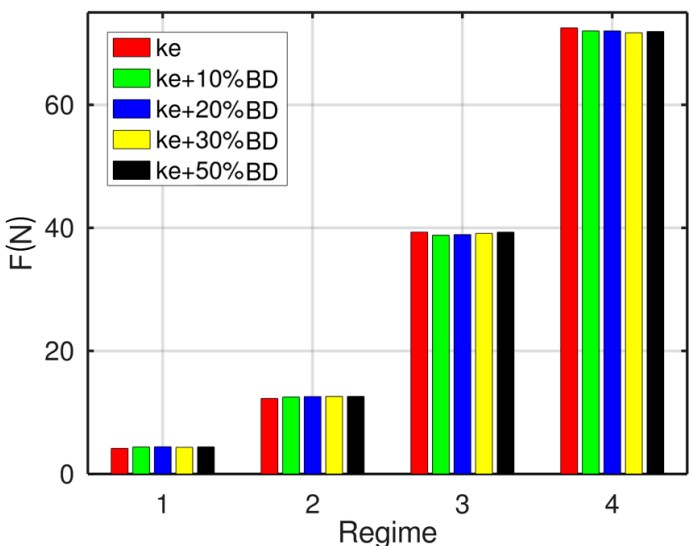

**Figure 11.** Variation of thrust F(N) depending on the regime and blend.

In all cases, the thrust of the turbo engine exhibits an increase when biodiesel concentration increases in the fuel blends tested, for all four studied operating regimes.

In Figure 12, the measured RMS (Root Mean Square) values for both the axial and radial vibrations are shown.

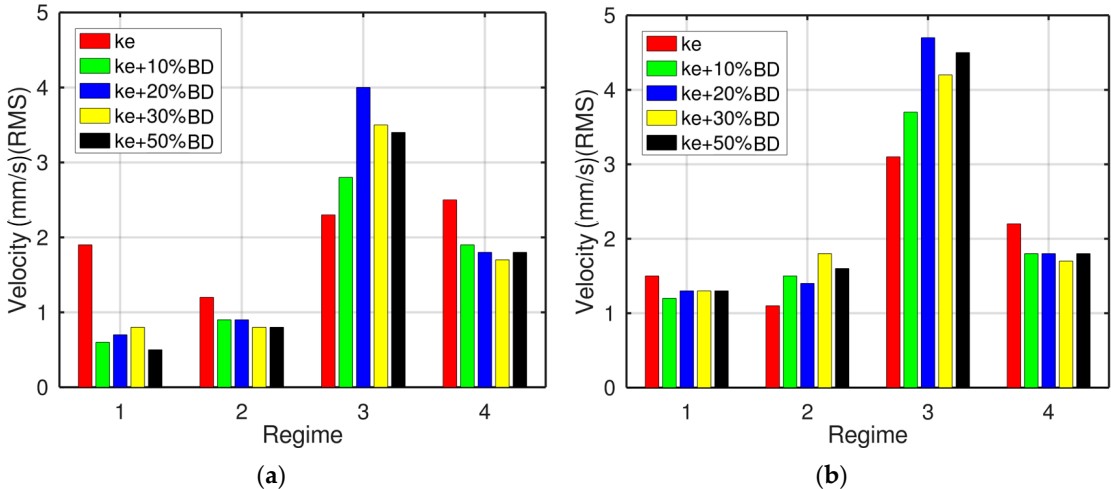

**Figure 12.** Vibration levels axial (**a**) and radial (**b**).

In the idle regime, the increase of the biodiesel concentration in the fuel blends resulted in a decrease of the vibrations level, while in the cruise operating regime, an increase of the vibrations level can be observed along with the increase of the biodiesel concentration in the fuel blends tested. However, all these variations do not threaten the stability and integrity of the turbo engine. In regime 3, it can be observed that the vibrations level increases along with the increase of the biodiesel concentration in the fuel blends tested. In the maximum operating mode, the vibrations level is slightly lower when the turbo engine is powered by the tested fuel blends in comparison with the turbo engine when powered by a Jet A type fuel.

It can be observed that, on the radial direction, the vibrations level exhibits lower values for all the tested fuel blends in the idle, cruise, and max operating regimes. In the case of operating regime 3, the vibrations level observed was higher when the turbo engine was powered by the fuel blends than when the turbo engine was powered by a Jet A type fuel.

Analysing Figure 12, it can be observed that, for operating regime 3, the vibrations are higher for all fuel blends tested including Jet A type fuel, compared with the other operating regimes. In order to investigate what happened when the turbo engine functioned at this speed, a vibration diagram was made and is presented in Figure 13.

An increase of $T_3$ was observed while the engine was operating in the idle regime, due to the increase of biodiesel concentration, while the vibration level decreased. These variations of the vibration levels may be influenced by the processes occurring in the combustion chamber, where the combustion seems to show some stability fluctuations during operation and, consequently, an effect on the variations of the vibrations on the axial direction.

The burning stability can be affected by using fuels other than Jet A fuel, causing vibration on the turbine and shaft, which can lead to engine breakdown. So, the vibration monitoring was used as a technique to achieve an image of the burning stability. It can be noticed that, for all studied cases, other than Jet A fuel, the vibrations fit the limits of functionality, with some regimes having slightly higher vibration levels. Therefore, the engine was not endangered during operating in different regimes.

Figure 13 presents the engine operation at different speeds. The entire test had a duration of 220 s, in which the engine operated from the lowest speed to the maximum speed. In the first 40 s, the engine was started; from 80 to 140 s, a run-up test was performed with a speed step of 1.25 k RPM/sec; followed by 20 s in maximum regime and then a rundown test. The vibration level highlights that, in the first 30 s, the engine automation rapidly increases the speed to pass the critical speeds, which produce vibration peaks especially on the Y direction with a maximum amplitude of 11 mm/s. Other resonances are observed from 90 to 120 s, which are mirrored in the rundown regime and occur at the same speeds

as in run up test. In can be noticed that, at a speed of around 80 k RPM, the vibration levels rise on both run up and run down. At maximum speed, the engine is stable, having low vibration on both axes.

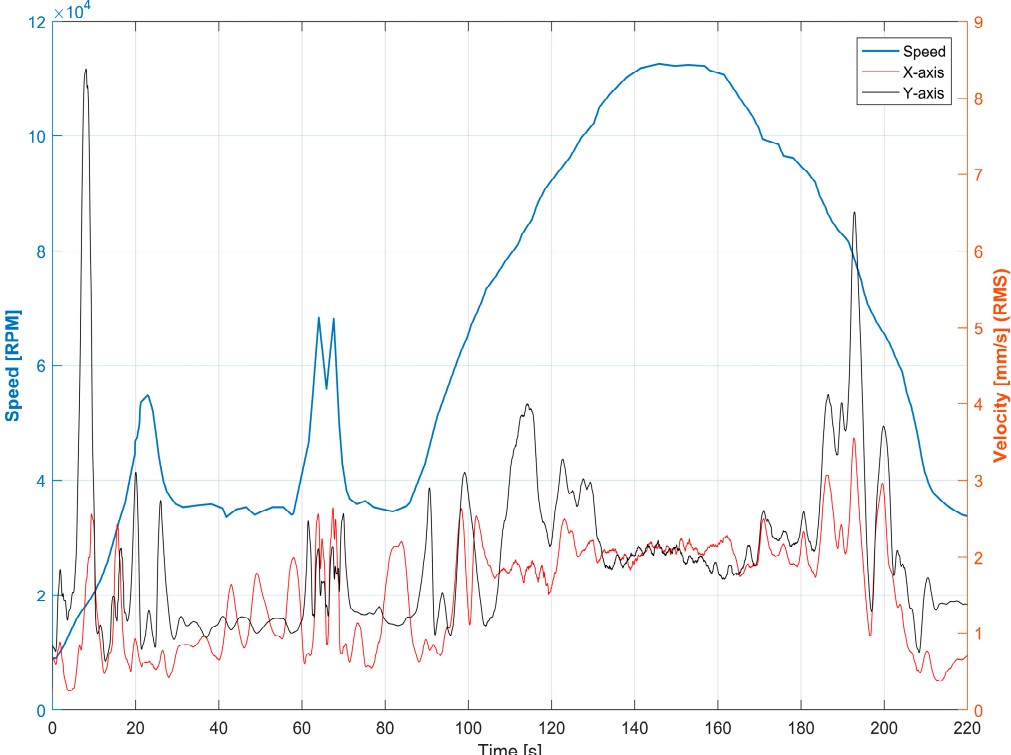

**Figure 13.** Vibration levels variation depending on the turbo engine speed.

The following section presents the spectral analysis of the sound signals for both microphones and for all tested blends, only for regime 3 (intermediate), this is depicted in Figure 14. This regime was chosen to be analysed from the vibration and noise point of view as significant differences were observed between the blends. These analyses are focused on the low frequencies where amplitudes differences are noticed, while at high frequencies, the amplitude differences between blends are insignificant, so the analysis will not take into account this frequency domain.

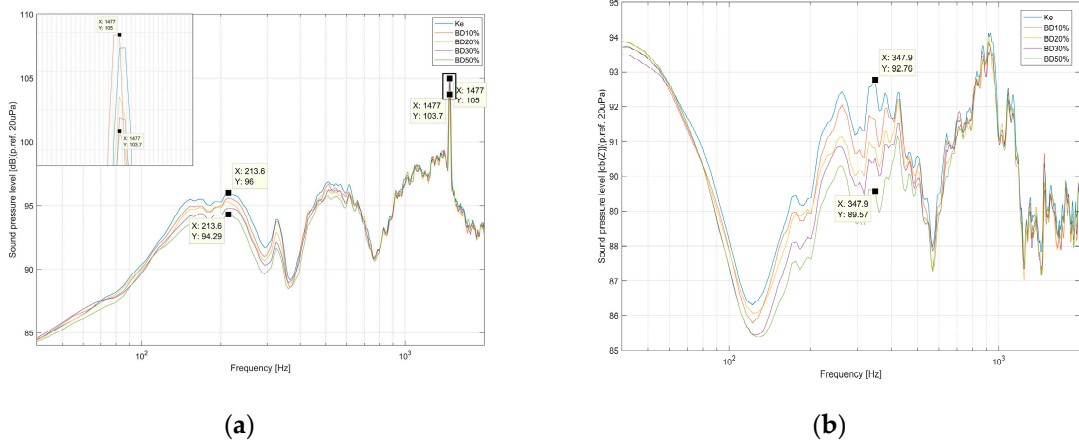

(**a**)                                (**b**)

**Figure 14.** Noise–FFT (Fast Fourier Transformation) time domain analysis for all blends, regime 3 ((**a**) Mic 1, (**b**) Mic 2).

The spectral analysis from Mic 1 highlights that, at low frequencies of around 213 Hz, the increase of the biodiesel concentration in fuel increases the noise reduction of around 2 dB. At these frequencies,

the noise is generated by both the aero-acoustic source of the jet flow and the thermo-acoustic source produced by the temperature variations known as entropy waves.

Because of the fact that the total mass flow was smaller during biodiesel blend tests, the noise differences are not caused by the jet gases. The differences came from the $T_3$ parameter, where the increase of biodiesel concentration leads to a decrease of the temperature of around 16.5 °C. Thus, the low frequencies noise reduction results from this temperature decrease, which leads to a smaller amplitude of the entropy waves. The same phenomenon is found in the other microphone, where the noise difference is around 3.2 dB. This is depicted in Figure 15.

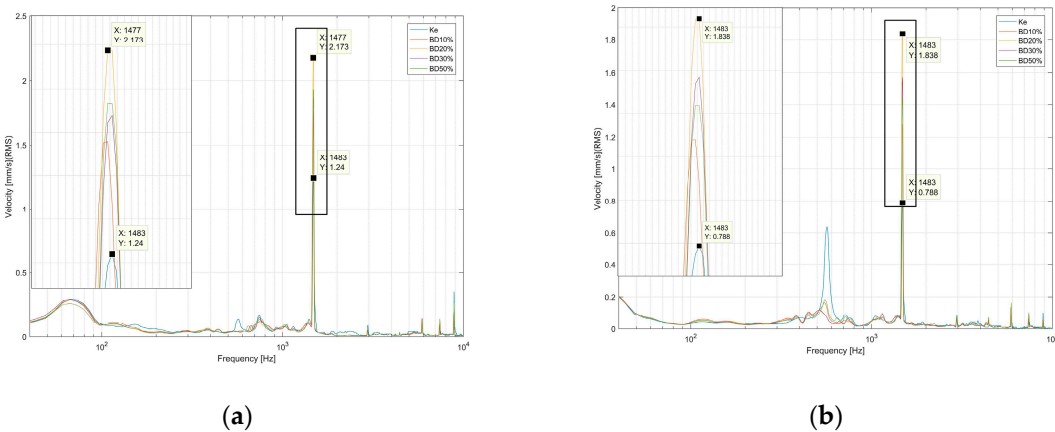

**Figure 15.** Vibration–FFT time domain analysis for all blends, regime 3 ((**a**) X; (**b**) Y).

Regarding the vibrations levels as a general remark, a higher biodiesel concertation leads to higher overall vibration levels. The spectral analysis of the vibration signal indicates that the main vibration source is represented by the speed component, and the vibration levels are within operating limits for all tested blends. It is interesting that the addition of the biodiesel led to doubling of the fundamental amplitude, which is quite strange because the fundamental is strictly related to the mechanic part, more specifically by the rotor unbalance. Thus, by adding biodiesel, it is expected to influence the thermodynamic regime of the engine and not the mechanic response of the engine parts. A possible explanation is that the burning of the biodiesel was causing a different temperature/pressure distribution on the turbine, which produced an apparent unbalance. In Alisaraei A. et. al. [63], a vibration analysis performed on a diesel engine that was tested using different percentages of biodiesel is presented. It was concluded that the vibration levels of different fuel blends do not follow a rule and depend on various parameters of the blends like the viscosity, lubrification properties, and flash points. Another remark was related to the higher viscosity of biodiesel blends and its influence on injection and powder in the injector nozzle, which is not performed properly.

From Table 1, it is observed that a higher concentration of biodiesel in fuel will lead to an increase of viscosity. Considering that our turboengine uses the fuel as lubricant for turbine bearing, this increase of vibration levels can be also explained by this fact.

## 7. Jet Engine Cycle Analysis

This section presents the performance variation for regime 4 (maximum regime). The foundation of the performance parameters computation is the cycle analysis of gas turbines, which was proven in [64].

$$F_{st} = \frac{F}{\dot{Mc} + \dot{Ma}} \left[ \frac{m}{s} \right] \tag{1}$$

where the $F_{st}$ is the specific thrust, F is the thrust force, $\dot{M}c$ is the fuel flow, and the $\dot{M}a$ is the air flow.

$$f = \frac{\dot{M}c}{\dot{M}a} = \frac{1}{LCV}\left(cp_3 \cdot T_3 - cp_2 \cdot T_2\right) \tag{2}$$

where f is the fuel–air ratio, LCV is lower calorific value, specific heat capacity, and $T_2$ is the temperature in front of the combustion chamber.

$$S = 3600 \cdot \frac{f}{F_{st}}\left[\frac{kg}{N \cdot h}\right] \tag{3}$$

where S is the specific fuel consumption. To determine performances obtained in the maximum regime of the turbo engine, the following calculations were made.

In the combustion chamber, there are concerns with respect to the incomplete combustion of the fuel, hence the combustion efficiency $\eta_b$ is expressed as follows:

$$\eta_b = \frac{\left(\dot{M}c + \dot{M}a\right)cp_3 \cdot T_3 - \dot{M}a \cdot cp_2 \cdot T_2}{\dot{M}a \cdot LCV} = \frac{(1+f)}{f \cdot LCV}\left(cp_3 \cdot T_3 - cp_2 \cdot T_2\right) \tag{4}$$

The thermal efficiency of an engine is another performance parameter that proved to be very useful in many cases. Thermal efficiency is defined as the net rate of organized out of the engine divided by the rate of thermal energy available from the fuel in the engine.

The thermal efficiency of the engine can be written as follows:

$$\eta_T = \frac{(1+f) \cdot v_e^2}{2 \cdot f \cdot LCV} = \frac{(1+f) \cdot F_{st}^2}{2 \cdot f \cdot LCV} \tag{5}$$

where $V_e$ is the gas flow velocity from the exhaust nozzle.

On the basis of the measured density, the fuel flow was converted from litters/hour in kg/second and, further on, the computation of the specific consumption can be done by means of Equation (3).

Table 3 presents the performance calculations for the maximum regime.

Analysing Table 3, it can be observed that the value of the thermal efficiency is very low, in contrast to the values from the literature, because operating procedures of a turbo engine differ from those of airplanes' turbo engines.

**Table 3.** Calculated performances obtained at the maximum regime for all tested blends.

| Fuel | $\eta_b$ (%) | $\eta_T$ (%) | S (kg/Nh) |
|------|------|------|------|
| Ke | 0.805 | 0.0516 | 0.0264 |
| Ke + 10%BD | 0.805 | 0.0518 | 0.0266 |
| Ke + 20%BD | 0.798 | 0.0514 | 0.0269 |
| Ke + 30%BD | 0.802 | 0.0520 | 0.0275 |
| Ke + 50%BD | 0.785 | 0.0516 | 0.0277 |

The burning efficiency and thermal efficiency exhibit small variations between the four tested combustible blends due to the increasing concentration of the biodiesel in the fuel blends tested.

From Figure 16, an increase of the specific consumption of fuel is observed as a result of the higher concentration of biodiesel in the combustible blends and a lower calorific power of the blend.

It can be observed that the specific consumption of the turbo engine shows small growth, which is normal considering that the calorific power of the biodiesel is lower than that of Jet A. Owing to the lower LCV and greater specific consumption of the fuel blends compared with Jet A, the introduction of biodiesel in aviation as fuel will lead to the necessity of larger sized fuel tanks.

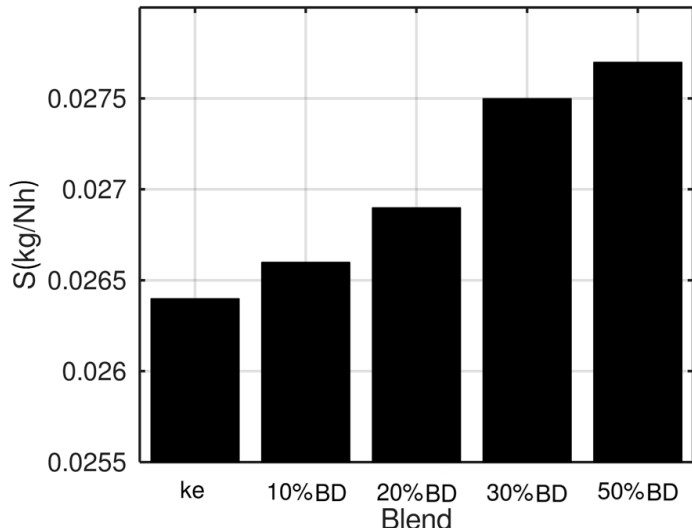

**Figure 16.** Variation of specific consumption for all tested blends at the maximum regime.

## 8. Conclusions

The measurements done on the Jet CAT P80® turbo engine show that the addition of the biodiesel in fuel does not endanger the functionality of the turbo engines.

A higher biodiesel concentration in blends will increase the freezing point, which leads to the impossibility of using these blends at high altitudes without being heated. The calorific value decreases with the growth of the biodiesel concentration, causing the increase of the specific consumption.

The combustion temperatures in front of the turbines grow with the increase of biodiesel concentration without endangering the engine integrity. The combustion efficiency and the thermal efficiency of the engine do not show significant variations between the kerosene and other mixtures.

The tests results presented in this paper showed that, for all the studied cases, other than Jet A fuel, the vibrations fit the limits of functionality, with some regimes having slightly higher vibration levels. On the third regime presented, at a speed of around 80 k RPM, the vibration levels are higher for the biodiesel blends than that of Jet A fuel. An explanation would be that the burning of biodiesel causes a different temperature/pressure distribution on the turbine, which produces an apparent imbalance.

The overall noise levels between the blends and Jet A are insignificant, but an interesting aspect was found at low frequencies in both noise signals. In both microphones, at low frequencies of around 213 Hz, the increase of the biodiesel concentration in fuel increases the noise reduction of around 2 dB and 3.2 dB. One assumption is that this phenomenon is caused by the smaller gas exhaust temperature, which generates smaller entropy noise.

**Author Contributions:** Conceptualization, G.C. and T.D.; software, G.C. and M.D.; investigation, G.C., M.D., R.M., L.G. and M.C.; data curation, G.C., R.M., L.C. and M.C.; formal analysis, G.C., M.D., R.M., L.C. and M.C.; writing original draft, G.C. and L.C.; supervision, G.C. and T.D.; methodology, T.D.; funding acquisition, G.C. and R.M.; writing, review and editing, R.M.; validation, M.D. All authors have read and agreed to the published version of the manuscript.

**Funding:** This research was founded by Executive Agency for Higher Education, Research, Development and Innovation Funding, grant no. 7N/2018 PN 18.10.01.05, And the APC was funded by Polytechnic University of Bucharest.

**Conflicts of Interest:** The authors declare no conflict of interest.

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
