# Peer review of "Investigating the Use of Recycled Pork Fat-Based Biodiesel in Aviation Turbo Engines"

_processes, doi:10.3390/pr8091196_

Round 1
Reviewer 1 Report
The article is very good and I basically just have these comments.
Axis labels and data are not readable in Figures 14 and 15. There is a lack of discussion and comparison of results with other authors.
After improving them, I recommend publishing the article.
Author Response
Point 1: Axis labels and data are not readable in Figures 14 and 15. There is a lack of discussion and comparison of results with other authors..
Response 1
The figures14 and 15 were restored.
In Alisaraei A. et. al.[58] paper it is presented a vibration analysis performed on a diesel engine that was tested using different percentages of biodiesel. It was concluded that the vibration levels of different fuel blends do not follow a rule and it depend to various parameters of the blends like the viscosity, lubrification properties and flash points. Another remark was related by the higher viscosity of biodiesel blends and its influence on injection and powder in injector nozzle which is not performed properly.
From the Table 1 it is observed that a higher concentration of biodiesel in fuel will lead to an increase of viscosity. Considering that our turboengine uses the fuel as lubricant for turbine bearing, this increase of vibration levels can be explained also due to this fact.

Reviewer 2 Report
- Line 42 subscript missing.
- Line 48 missing space between citation.
- Line 50 only one reference for such statement??
- Line 54 and 59 it is a big understatement to refer to such a small number of references with such a broad and well-described topic.
- Line 93 what is a mixture of kerosene Jet A (Ke) with 100% of biodiesel?? The sentence must be write differently.
- Line 116 missing space between citation.
- Line 121 missing space between citation.
- Line 145 missing full stop.
- Line 151 why blends in this sentence are different from that mentioned in line 93?
- Line 226 the title left on the previous page.
- Figure 2 – 6 are not mentioned in the text.
- Figure 8 low quality.
- Figure 13 very low quality. It must be changed.
- Figure 14 very low quality. It must be changed.
- Figure 15 very low quality. It must be changed.
- Conclusions should be more specific and may contain plans of authors in this field of interest.
Author Response
Point 1 Line 42 subscript missing.
Response 1
Done
Point 2 Line 48 missing space between citation.
Response 2
Done
Point 3 Line 50 only one reference for such statement??
Response 3
We consider that the reference presented contains sufficient information. If it is required, we will add other relevant references.
Point 4 Line 54 and 59 it is a big understatement to refer to such a small number of references with such a broad and well-described topic.
Response 4
The paper does not focus on the engine piston and we consider that the references contain inside sufficient information. If it is necessary, we will add additional relevant references.
Point 5 Line 93 what is a mixture of kerosene Jet A (Ke) with 100% of biodiesel?? The sentence must be write differently.
Response 5
Done
Point 6 Line 116 missing space between citation.
Response 6
Done
Point 7 Line 121 missing space between citation.
Response 7
Done
Point 8 Line 145 missing full stop.
Response 8
Done
Point 9 Line 151 why blends in this sentence are different from that mentioned in line 93?
Response 9
There was a mistake. Done.
Point 10 Line 226 the title left on the previous page.
Response 10
Point 11 Figure 2 – 6 are not mentioned in the text.
Response 11
Indeed, Done
Point 12 Figure 8 low quality.
Response 12
Done
Point 13 Figure 13 very low quality. It must be changed.
Response 13
Done
Point 14 Figure 14 very low quality. It must be changed.
Response 14
Done
Point 15 Figure 15 very low quality. It must be changed.
Response 15
Done
Point 16
Conclusions should be more specific and may contain plans of authors in this field of interest.
Response 16
The future plans include testing biodiesel produced from other types of fat or other types of biodiesel including the monitoring of emissions and the studies will be executed on turbo engines of bigger dimensions found in the testing stand of turbo engines that belongs to COMOTI.

Reviewer 3 Report
The authors present their work on investigating the use of recycled pork fat-based biodiesel in aviation turbo engines. The work is of interest to the researches in this field as it lays the groundwork for similar investigations all leading toward making air-travel and transportation more sustainable. The following are my comments:
- A freezing point of -40 C for Jet A fuel and -47 C for Jet A1 fuel is required. Table 1 shows the evaluation metric to be less than -35 C. The authors should justify why -40 C or -47 C was not used as a standard.
- Page 4, line 151: A justification/rationale for the choice of experimental space design (10%, 20%, 30%, 50%) should be included
- The conclusions are relevant to the study. The authors should include comments on potential future work
Author Response
Point 1:. A freezing point of -40 C for Jet A fuel and -47 C for Jet A1 fuel is required. Table 1 shows the evaluation metric to be less than -35 C. The authors should justify why -40 C or -47 C was not used as a standard.
Response 1
This temperature from table 1 was presented as the measuring equipment available in the laboratory cannot measure the tempetatures of -40 or -47 degrees.
Point 2:. Page 4, line 151: A justification/rationale for the choice of experimental space design (10%, 20%, 30%, 50%) should be included
Response 2
This concentrations have been chosen having in focus the safety conditions for the turbo engine and in line with the available references.
Point 3:. The conclusions are relevant to the study. The authors should include comments on potential future work
Response 3
The future plans include testing biodiesel produced from other types of fat or other types of biodiesel including the monitoring of emissions and the studies will be executed on turbo engines of bigger dimensions found in the testing stand of turbo engines that belongs to COMOTI.
